# Accuracy of a deep convolutional neural network in the detection of myopic macular diseases using swept-source optical coherence tomography

Takahiro Sogawa[1], Hitoshi Tabuchi[1,2], Daisuke Nagasato[1,2]*, Hiroki Masumoto[1,2], Yasushi Ikuno[3], Hideharu Ohsugi[4], Naofumi Ishitobi[1], Yoshinori Mitamura[5]

**1** Department of Ophthalmology, Tsukazaki Hospital, Himeji, Japan, **2** Department of Technology and Design Thinking for Medicine, Hiroshima University Graduate School, Hiroshima, Japan, **3** Ikuno Eye Center, Osaka, Japan, **4** Ohsugi Eye Clinic, Kobe, Japan, **5** Department of Ophthalmology, Institute of Biomedical Sciences, Tokushima University Graduate School, Tokushima, Japan

* d.nagasato@tsukazaki-eye.net

## Abstract

This study examined and compared outcomes of deep learning (DL) in identifying swept-source optical coherence tomography (OCT) images without myopic macular lesions [i.e., no high myopia (nHM) vs. high myopia (HM)], and OCT images with myopic macular lesions [e.g., myopic choroidal neovascularization (mCNV) and retinoschisis (RS)]. A total of 910 SS-OCT images were included in the study as follows and analyzed by k-fold cross-validation (k = 5) using DL's renowned model, Visual Geometry Group-16: nHM, 146 images; HM, 531 images; mCNV, 122 images; and RS, 111 images (n = 910). The binary classification of OCT images with or without myopic macular lesions; the binary classification of HM images and images with myopic macular lesions (i.e., mCNV and RS images); and the ternary classification of HM, mCNV, and RS images were examined. Additionally, sensitivity, specificity, and the area under the curve (AUC) for the binary classifications as well as the correct answer rate for ternary classification were examined.

The classification results of OCT images with or without myopic macular lesions were as follows: AUC, 0.970; sensitivity, 90.6%; specificity, 94.2%. The classification results of HM images and images with myopic macular lesions were as follows: AUC, 1.000; sensitivity, 100.0%; specificity, 100.0%. The correct answer rate in the ternary classification of HM images, mCNV images, and RS images were as follows: HM images, 96.5%; mCNV images, 77.9%; and RS, 67.6% with mean, 88.9%. Using noninvasive, easy-to-obtain swept-source OCT images, the DL model was able to classify OCT images without myopic macular lesions and OCT images with myopic macular lesions such as mCNV and RS with high accuracy. The study results suggest the possibility of conducting highly accurate screening of ocular diseases using artificial intelligence, which may improve the prevention of blindness and reduce workloads for ophthalmologists.

**Data Availability Statement:** All relevant data are within the paper and its Supporting Information files.

**Funding:** The authors received no specific funding for this work.

**Competing interests:** The authors have declared that no competing interests exist.

# Introduction

Myopia is a kind of refractive error wherein an image is formed in front of the retina due to increases in axial length and refractive power, regardless of the intensity of the error and age of onset [1]. Myopia is associated with macular complications such as myopic choroidal neovascularization (mCNV), retinoschisis (RS), and myopic chorioretinal atrophy, which can lead to blindness. Recently, the prevalence of myopia has been increasing annually around the world, especially in East Asia, and vision loss caused by myopia is considered a global social problem [2–6].

Traditionally, the evaluation of the retina has largely been conducted by ophthalmoscope. However, this device only observes the retina directly, making the completion of an objective evaluation difficult. Optical coherence tomography (OCT) has recently made it possible to obtain detailed tomographic images of the retina noninvasively and in a small amount of time. Swept-source OCT (SS-OCT), in particular, can capture high-quality images using a light source with deep penetration into the tissue, arithmetic mean, and tracking function of the fundus [7–9]. With the advancement of such OCT technology, research on myopic macular diseases such as RS and mCNV, which are directly related to decreased visual function, has progressed dramatically. Studies using OCT have shown that early surgical intervention is important for the maintenance of long-term visual function in the context of RS [10–15] and that early anti–vascular endothelial growth factor drug administration can help to maintain long-term visual function in mCNV [16–20]. Therefore, early detection and treatment of macular lesions associated with myopia are crucial to maintaining better vision. However, administering screening tests to all people with myopia is not realistic from the human resource or economic perspective [21].

In recent years, artificial intelligence (AI) technologies, including deep learning (DL), have made remarkable progress and, in the medical field, various applications in diagnostic imaging have been reported [22]. In the field of ophthalmology, many researchers, including the authors, have already reported applications of DL to image analysis using OCT, OCT angiography, and ultrawide-field fundus ophthalmoscopy [23–33].

To the best of our knowledge, however, there have been no studies performed involving the automatic diagnosis of myopic macular disease using DL technology for SS-OCT images. If AI can establish diagnoses as accurately as ophthalmologists can using DL, such would significantly contribute to the early detection of myopia-related complications, which may help decrease the number of patients who would suffer from loss of vision.

In light of the above, this study sought to examine and compare DL's classification performance using OCT images without myopic macular lesions [i.e., no high myopia (nHM) vs. high myopia (HM)] and OCT images with myopic macular lesions (e.g., mCNV and RS).

# Materials and methods

## Image dataset

A total of 910 SS-OCT images of HM with normal eyes or myopic macular lesions were included in our study; images with reduced clarity of the eye due to severe cataract and/or severe vitreous hemorrhage were excluded. All images were taken using SS-OCT (Topcon DRI OCT-1 Atlantis; Topcon Corp., Tokyo, Japan). Horizontal scans (12 mm) on the fovea were performed by trained certified orthoptists. nHM was defined as having an axial length of less than 26 mm, while HM was defined having an axial length of 26 mm or more and with neither involving other obvious ocular diseases. The purpose of this study was to evaluate the DL's ability to detect a single condition; therefore, in cases with mCNV or RS with myopic macular

lesions, images showing complications of other retinal diseases (e.g., diabetic retinopathy, retinal vein occlusion) were also excluded. These SS-OCT images were classified into nHM, HM, mCNV, and RS by retinal specialists. Some nHM and HM images included with comorbidities (mild cat, chorioretinal atrophy, epiretinal membrane, macular hole and so on). Some RS images included retinoschisis with retinal detachment. Representative images of each class are presented in Fig 1.

The obtained images were trained and validated using k-fold cross-validation (k = 5). With this approach, image data were split into k groups, and (k − 1) groups were used as training data, while one group was used for validation [34,35]. The process was repeated k times until each of the k groups reached the validation dataset.

Data augmentation techniques, including brightness, gamma correction, histogram equalization, noise addition, and inversion, were applied to the images in the training dataset to increase the amount of training data by sixfold. Then, deep neural network (DNN) models were constructed and trained using the preprocessed image data.

Because of the retrospective and observational nature of the study, the need for written informed consent was waived by the ethics committees. The data acquired in the course of the data analysis were anonymized before we accessed them. This study was conducted in compliance with the principles of the Declaration of Helsinki and was approved by the local ethics committees of Tsukazaki Hospital,

## Deep-learning model and training

In this study, the following nine DNN models were constructed and trained: Visual Geometry Group-16 (VGG16), Visual Geometry Group-19 (VGG19), Residual Network-50 (ResNet50), InceptionV3, InceptionResNetV2, Xception, DenseNet121, DenseNet169, and DenseNet201. After the training, the performance of each model was evaluated using test data [36–38].

The convolutional DNN automatically learns local features of images and identifies images based on said information [39–41]. Among the nine DNN models used in this study, the network architecture of a well-known model, VGG16, in particular is explained (Fig 2). The original SS-OCT image size was 1,038 × 802 pixels but it was resized to 256 × 192 pixels to shorten the analysis time. The images were read as color images; the size of the input tensor was 256 × 192 × 3. Since each pixel value was in the range of zero to 255 pixels, it the value was first divided by 255 and normalized according to the range of zero to one pixel(s).

The VGG16 model consists of five convolutional blocks and some fully connected layers. Each block involves a convolutional layer and a maximum pooling layer. A convolutional layer is a block that captures features in images. Since the stride was set to 1, the downsizing of images was not performed in the convolutional layer. The Rectified Linear Unit (ReLU) activation function was used to solve the vanishing gradient problem [42]. Additionally, the Max-Pooling layer's stride was set to 2, and the images were downsized to compress the information [43].

Next, after passing through five blocks, a flattened layer and two fully connected layers were realized. The flattened layer deletes position information from the tensor representing the features extracted by the convolutional block, while the fully connected layers compress the information received from the previous layers and pass it on to the next layer. The softmax function produce the probability of each class was deemed as the final output.

Fine-tuning was applied to increase the learning speed to achieve high performance with limited data [44]. The parameters obtained by learning ImageNet were used as initial parameters of the convolutional layer blocks.

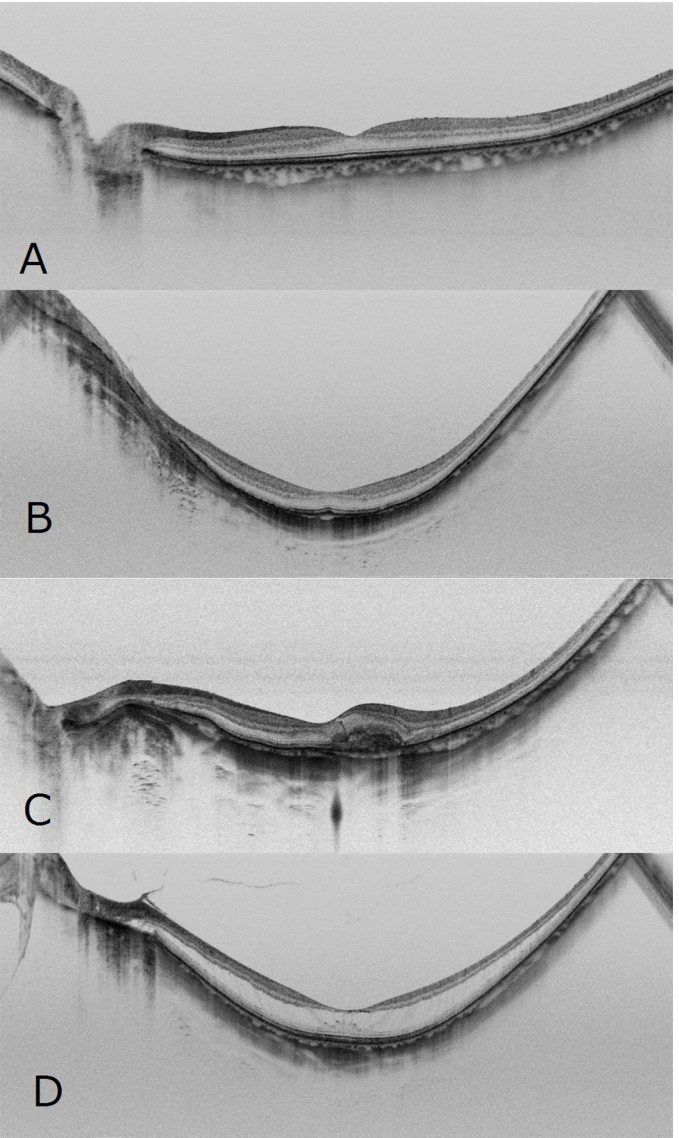

**Fig 1. Representative horizontal scans of SS-OCT.** Normal OCT image without HM (A), OCT image with HM and no macular lesions (B), and OCT images with mCNV (C) and RS (D) of the left eye using SS-OCT.

For weights and biases, a Stochastic Gradient Descent (learning rate = 0.0001, momentum term = 0.9) was used as an optimizer to update the parameters [45,46]. The code used to perform the above is shown in S1 File.

An ensemble model was constructed by averaging the output of any network type among the nine network types. Thus, $2^9-1$ types of ensemble models were constructed. Among them, the classification performance of the model with the highest AUC for binary classification and that of the model with the highest overall correct answer rate for ternary classification were evaluated and compared with those abilities of human ophthalmologists, described later.

The models were constructed and evaluated using Python Keras (https://keras.io/ja/) [Backend is TensorFlow (https://www.tensorflow.org/)]. For development and validation, the following computing setup was used: Intel Core i7-7700K® (Intel, Santa Clara, CA, USA) as the

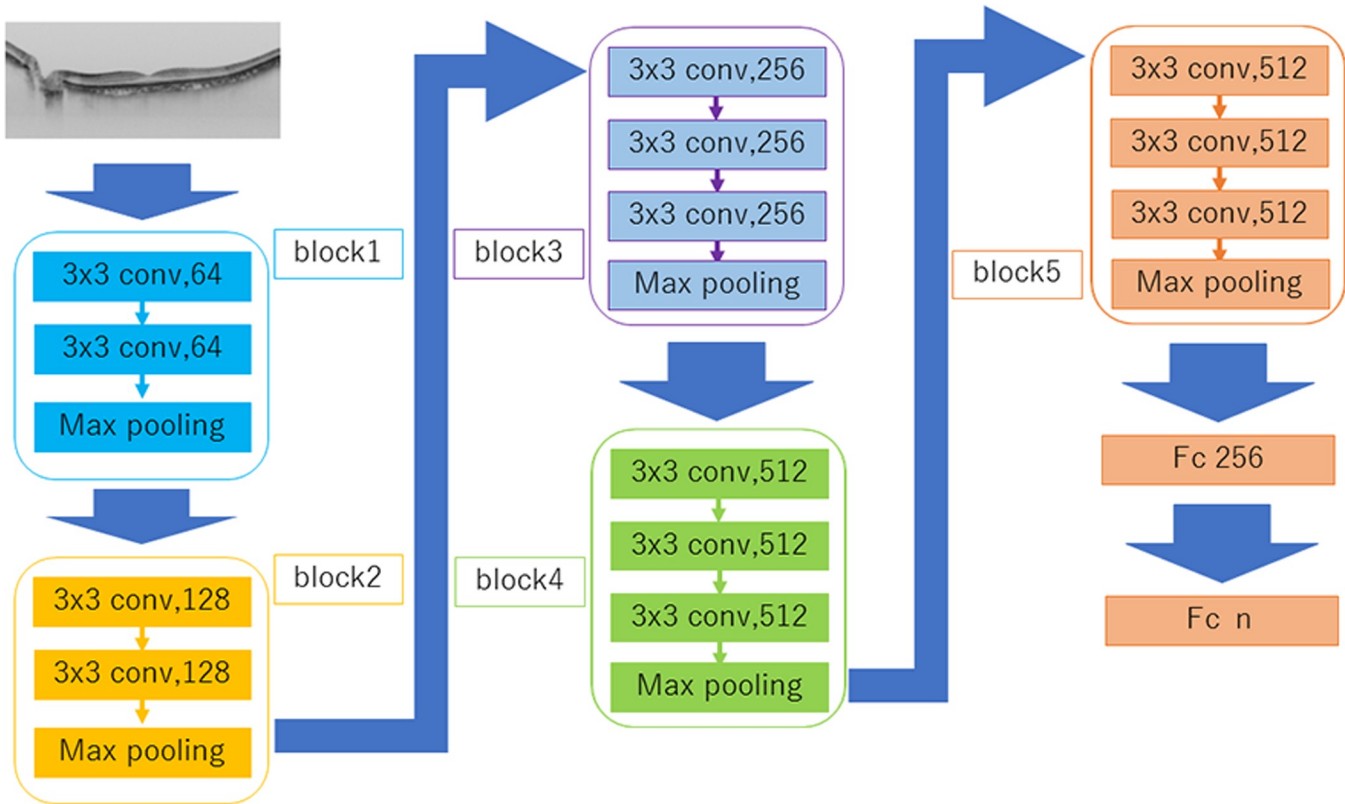

**Fig 2. Overall architecture of the VGG16 model.** The image data were converted to a pixel resolution of 256 × 192 pixels and set as the input tensor. After placing the convolution layers (Conv 1, 2, and 3), activation function (ReLU), pooling layers (MP 1 and 2) after Conv 1 and 3 and a dropout layer (drop rate: 0.25) all were passed through two fully connected layers (FC 1 and 2). In the final output layer, the classification was performed using some class softmax functions.

central processing unit and NVIDIA GeForce GTX 1080 (Nvidia, Santa Clara, CA, USA) as the graphics processing unit.

## Comparison with ophthalmologists

For the binary classification of OCT images with or without myopic macular lesions, 46 OCT images without myopic macular lesions (nHM: 23 images; HM: 23 images) and 46 OCT images with myopic macular lesions (mCNV: 23 images; RS: 23 images) were included. For the binary classification of HM images and images with myopic macular lesions (mCNV images and RS images), 44 images of HM and 44 images of myopic macular disease (mCNV: 22 images; RS: 22 images) were included. In addition, for the ternary classification of HM images, mCNV images, and RS images, 23 images of each were included. The task of classifying these images was given to three human ophthalmologists and their results were compared with those of the neural networks. The metrics used to evaluate the classification performance of the neural networks and ophthalmologists were the AUC for the binary classifications and overall accuracy for the ternary classification, respectively.

## Outcome

Performance results of the binary classification of OCT images with or without myopic macular lesions, the binary classification of HM images and images with myopic macular lesions

(mCNV images and RS images), and the ternary classification of HM images, mCNV images, and RS images were examined. Outcome measures for the binary classifications were AUC, sensitivity, and specificity, obtained from the receiver operating characteristic (ROC) curve. Based on the probability value output by the neural network as an abnormal group, the ROC curve was obtained by changing the threshold value to pass judgment regarding whether or not they were myopic macular disease images. For outcomes of the ternary classification, among the three diagnostic possibilities output by the neural network, the maximum value was used for diagnosis. The overall accuracy and accuracy within each group were obtained by comparing the diagnosis given by the network and the actual diagnosis from the ophthalmologists.

## Heatmap

A gradient-weighted class activation mapping (Grad-CAM) method [47] was used to create heatmap images that indicated where the DNN was focused. As an example, each heatmap in the VGG16 network is shown in Fig 3. The output of the second convolutional layer of the second convolutional block was maximized, and the Grad-CAM method was used. The ReLU function was employed to correct the loss function during backpropagation. This process was performed by Python Keras-Vis (https://raghakot.github.io/keras-vis/).

## Statistical analysis

In the comparison of subjects' demographic data, an analysis of variance was used for age and axial length. The chi-squared test was used for categorical variables (sex ratio and right:left ratio).The 95% confidence interval of the AUC was calculated using the following formula, assuming a normal distribution [48]:

$$95\%CI = A + Z(0.05/2) * SE(A)$$

$$Z(x) = \frac{1}{\sqrt{2\pi}} e^{-\frac{x^2}{2}}$$

$$SE(A) = \sqrt{\frac{A * (1 - A) + (n_p - 1) * (Q_1 - A^2) + (n_n - 1) * (Q_2 - A^2)}{n_P * n_N}}$$

$$Q_1 = \frac{A}{2 - A}$$

$$Q_2 = \frac{2A^2}{1 + A}$$

$n_P$  The amount of Good groups, (1) 563 (2) 456

  $n_N$  The amount of Normal images, (1) 233 (2) 233

  Sensitivity and specificity when the threshold to determine prevalence was set to 0.5 were obtained. The 95% CIs for sensitivity and specificity were calculated using the Clopper–Pearson method. The correct answer rate in the context of ternary classification was also obtained using the Clopper–Pearson method.

  A significant difference was determined when the p-value was less than 0.05 (p < 0.05). These statistical analyses were performed using Python SciPy (https://www.scipy.org/), Python

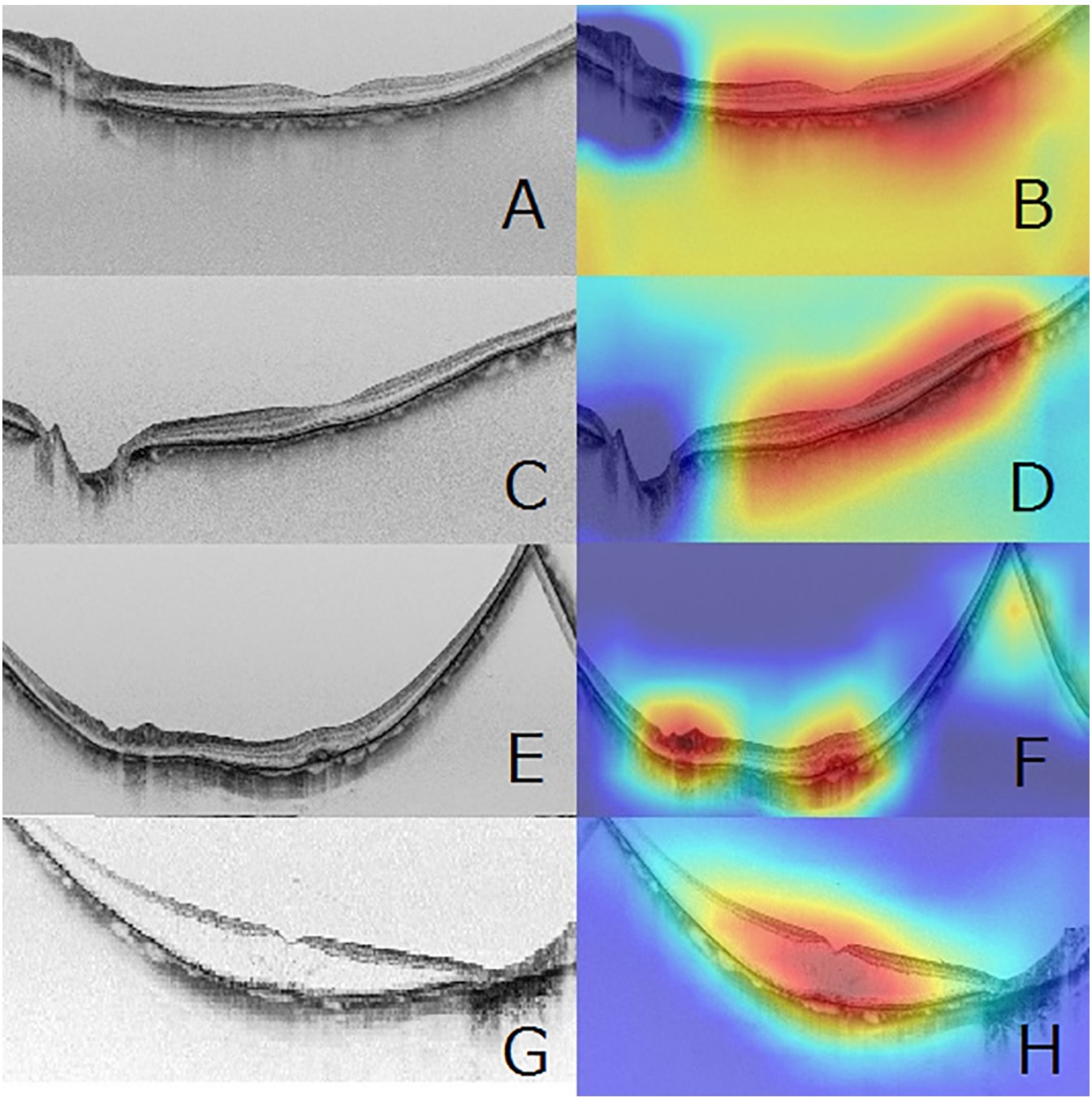

**Fig 3. Representative horizontal scans of SS-OCT and corresponding heatmaps.** Presented are a normal SS-OCT image with nHM (A), and its corresponding superimposed heatmap (B); OCT image with HM and no macular lesions (C) and its corresponding superimposed heatmap (D); OCT image with myopic choroidal neovascularization (E) and its corresponding superimposed heatmap (F); and OCT image with myopic retinoschisis (G) and its corresponding superimposed heatmap (H). For all of them, the convolutional DNN focused on the macular area (red color) on the SS-OCT images (B, D, F, and H). In particular, the DNN focused on the lesion area of the SS-OCT images in the images with retinoschisis and myopic choroidal neovascularization.

statsmodel (http://www.statsmodels.org/stable/index.html), and R pROC (https://cran.r-project.org/web/packages/pROC/pROC.pdf).

## Results

Table 1 shows demographic data of 910 subjects from whom 910 study images were obtained. There was no significant difference between the four groups in terms of the ratio of left and right eyes (p = 0.6585, chi-squared test); however, significant differences were found in age, sex, and axial length. (p < 0.001, p < 0.005, and p < 0.001, respectively; chi-squared test and analysis of variance).

### Neural network performance

For the binary classification of OCT images with or without myopic macular lesions, an ensemble model of VGG16, VGG19, DenseNet121, InceptionV3 and ResNet50 showed the best performance as follows: AUC, 0.970; sensitivity, 90.6%; and specificity, 94.2%.

For the binary classification of MH images and images with myopic macular lesions (mCNV and RS), VGG16 showed the best performance as follows: AUC, 1.000; sensitivity, 100.0%; and specificity, 100.0% (Table 2).

Finally, for the ternary classification of HM images, mCNV images, and RS images, VGG16 and DenseNet121 showed the best performance as follows: HM, 96.5%; mCNV, 77.9%; and RS, 67.6%. The overall correct answer rate was 88.9% (Table 3).

### Comparison of neural network and ophthalmologist outcomes

For the binary classification of a total of 92 OCT images with or without myopic macular lesions, the neural networks' performance was AUC: 0.837, whereas the ophthalmologists' performance was AUC: 0.877 (p = 0.86).

For the binary classification of a total of 88 HM images and images with myopic macular lesions (mCNV and RS), the neural networks' performance was AUC: 1.000, whereas the ophthalmologists' performance was AUC: 0.875 (p = 0.48).

Finally, for the ternary classification of a total of 69 images (HM, mCNV, and RS), the neural networks' performance for overall accuracy was 79.7%, whereas the ophthalmologists' performance for the same was 86.0% (p = 0.76).

In all three classifications, no significant difference was found between the results of neural networks and those of the ophthalmologists (Table 4).

### Heatmap

The corresponding heatmaps of the representative SS-OCT images of nHM, HM, mCNV, and RS are shown in Fig 3. In the heatmaps, red is used to indicate the strength of deep convolutional neural network focus.Increases in color intensity were observed around the macula in

**Table 1. Subject demographics.**

|  | nHM | HM | mCNV | RS | p-value |
|---|---|---|---|---|---|
| N | 146 | 531 | 122 | 111 | |
| Age (years) | 64.5 ± 13.5 | 58.3 ± 14.0 | 68.6 ± 9.3 | 64.7 ± 11.5 | <0.001* |
| Sex (female) | 73 (50.0%) | 356(67.0%) | 97 (79.5%) | 87 (78.4%) | <0.005** |
| Eye (left) | 76 (52.1%) | 273 (51.6%) | 71 (41.8%) | 57 (48.6%) | 0.66** |
| AL (mm) | 24.4 ± 1.3 | 28.1 ± 1.7 | 29.2 ± 1.7 | 29.4 ± 1.8 | <0.001* |

nHM, no high myopia; HM, high myopia; mCNV, myopic choroidal neovascularization; RS, retinoschisis; AL, axial length;

*analysis of variance,

**chi-squared test.

**Table 2. Results of the binary classifications.**

|  | nHM and HM vs. mCNV and RS | HM vs. mCNV and RS |
|---|---|---|
| AUC | 0.970 (0.939–1.000) | 1.000 (1.000–1.000) |
| Sensitivity | 90.6 (86.1–95.1) | 100.0 (98.3–100.0) |
| Specificity | 94.2 (92.1–96.3) | 100.0 (99.2–100.0) |

nHM, no high myopia; HM, high myopia; mCNV, myopic choroidal neovascularization; RS, retinoschisis; AUC, area under the curve.

95% CIs are presented in parentheses.

nHM and HM images, in the highlighted area due to choroidal neovascular at the macula in mCNV images, and in the RS area at the macula in RS images.

## Discussion

In this study, using the combination of nine DNN models including VGG16, VGG19, ResNet50, InceptionV3, InceptionResNetV2, Xception, DenseNet121, DenseNet169, and DenseNet201, the classification of myopic macular diseases (mCNV and RS) and no myopic macular disease was conducted using SS-OCT images. The results showed that our DL models was able to classify both no myopic macular disease and myopic macular diseases with high accuracy. The combination of DNN models provided a correct answer rate that was equivalent to that of the ophthalmologists for each classification. To our knowledge, this study is the first to report on the classification ability of DL with high accuracy among RS and mCNV images using SS-OCT images.

A few recent studies considering AI's detection ability using OCT images have been conducted on age-related macular degeneration (AMD). Treder et al. [49] developed and evaluated a DL program to detect AMD in spectral-domain OCT(SD-OCT). Their approach was tested using 100 OCT images (AMD: 50; healthy control: 50) and yielded correct answer rates of 0.997 in the AMD group and 0.920 in the healthy control group for a high level of significance (p < 0.001). Yoo et al. [50] also evaluated the automated detection of AMD in both OCT and fundus images using a DL program. Here, the DL with OCT images alone showed an AUC of 0.906 and a correct answer rate of 82.6%, the DL with fundus images alone presented an AUC of 0.914 and a correct answer rate of 83.5%, and the DL with a combination of OCT and fundus images showed an AUC of 0.969 and a correct answer rate of 90.5%. Similarly, AI's diagnostic efficiency in OCT images has been reported in correlation with other diseases. Our results concerning AI's diagnostic performance in images with myopic macular diseases also showed similar sensitivity and AUC outcomes as those in previous reports. A neural network can devise and construct an optimal structure to learn and detect local features of complex image data with individual differences [39,41,51].

In our study, we succeeded in obtaining diagnostic accuracy comparable to that of a human ophthalmologist by using the ensemble method, which combines various DL models as an AI

**Table 3. Results of the ternary classification.**

|  | HM | mCNV | RS | Average |
|---|---|---|---|---|
| Correct answer rate | 96.5 | 77.9 | 67.6 | 88.9 |

HM, high myopia; mCNV, myopic choroidal neovascularization; RS, retinoschisis.

Data are presented in %.

**Table 4. Results of the comparison between outcomes of neural networks and humans.**

|  | Neural networks | Ophthalmologists | p-value |
|---|---|---|---|
| nHM and HM vs. mCNV and RS | 0.837 (0.745–0.906) | 0.877 (0.832–0.913) | 0.86 |
| HM vs. mCNV and RS | 1.000 (0.959–1.000) | 0.875 (0.829–0.912) | 0.48 |
| Overall accuracy of HM, RS, and mCNV | 79.7 (68.3–88.4) | 86.0% (80.5–90.4) | 0.76 |

nHM, no high myopia; HM, high myopia; mCNV, myopic choroidal neovascularization; RS, retinoschisis.

95% CIs are presented in parentheses.

algorithm. In the classifications directed by AI, the lesion sites where AI actually detected the reported findings often differ from the essential lesion sites that ophthalmologists examine. However, in this study, heatmaps were used to show where the neural network focused, revealing increases in color intensity at the following sites: around the macula in nHM and HM SS-OCT images, at the RS site in RS images, and at the mCNV site in mCNV images. These focus sites match with the sites on which the ophthalmologists focus on during diagnosis, indicating that DNN accurately identified the locations of RS and mCNV lesion sites and classified between normal images and images of myopic macular diseases based on the features of the lesions. However, strictly speaking, it is difficult to compare the diagnostic performance between humans and AI. Liu et al. [52] conducted a systematic review and meta-analysis to compare the diagnostic accuracy of DL algorithms with that of health care providers using medical images. Among the previous studies they examined, there were 14 studies that compared DL models and health care providers using the same sample data and met other criteria, including the publication of raw data. When aggregating the performance data of these 14 studies, they found a mean sensitivity of 87.0% (95% CI: 83.0–90.2) and a mean specificity of 92.5% (95% CI: 85.1–96.4) for the DL models and a mean sensitivity of 86.4% (95% CI: 79.9–91.0) and a mean specificity of 90.5% (95% CI: 80.6–95.7) for health care providers. Their study therefore suggested that the diagnostic performance of DL models was equivalent to that of health care providers. However, they also pointed out the existing lack of quality studies comparing AI and medical professionals, with no established comparison method currently available for use. In the present study, the DL model we used was able to obtain the same correct answer rate relative to the ophthalmologists using the same sample data. However, there is still room for improvement in this type of research, including in the areas of increasing the number of images for training, further improving the AI algorithms, and combining OCT and fundus images.

At present, the early detection of myopic macular diseases requires an examination performed by an ophthalmologist, but there are not enough ophthalmologists worldwide to pursue this. Our study results found no significant difference in classification performance between the neural networks and ophthalmologists in the binary classification of OCT images with or without myopic macular lesions; in the binary classification of HM images and images with myopic macular lesions; or in the ternary classification of HM images, mCNV images, or RS images. This suggests that the conduct of automated diagnosis by AI using SS-OCT image data, which can be acquired noninvasively and easily, may be very useful in myopic macular disease screening.

Our study has a few limitations that should be considered. First, imaging diagnosis by AI would be impossible among patients with reduced clarity of the eye due to severe cataracts or severe vitreous hemorrhage and in patients for whom detailed imaging cannot be obtained due to severely poor fixation. For these reasons, such SS-OCT images were excluded from this study. Second, in this study, the demographic data varied between groups. Myopia is

significantly more common in females than in males and the prevalence of myopic macular diseases is significantly higher in older populations; therefore, the influence of such demographic data seems to be unavoidable [53–55]. Third, the AI algorithms created and tested herein might not be generelizable to other commercially available similar imaging devices, because we investigated by only the Topcon DRI OCT-1. Fourth, to shorten the analysis time, the original SS-OCT image with 1,038 × 802 pixels was resized to 256 × 192 pixels. Finally, mCNV and retinal hemorrhage showed similar findings in SS-OCT images. Therefore, data from sources other than OCT images are required to distinguish these conditions [56,57].

## Conclusion

The DL model was able to classify between myopic macular diseases (mCNV and RS) and no myopic macular disease with high accuracy using SS-OCT images. These findings suggest that DL is useful in reducing ophthalmologists' workloads in screening and preventing vision loss in myopic macular disease patients.

## Supporting information

**S1 File.**
(ZIP)

## Acknowledgments

We thank Masayuki Miki and the orthoptists at Tsukazaki Hospital for support in collecting the data.

## Author Contributions

**Conceptualization:** Hitoshi Tabuchi, Yasushi Ikuno.

**Data curation:** Hiroki Masumoto.

**Formal analysis:** Hiroki Masumoto.

**Investigation:** Naofumi Ishitobi.

**Methodology:** Hideharu Ohsugi.

**Project administration:** Daisuke Nagasato.

**Supervision:** Hitoshi Tabuchi.

**Validation:** Hiroki Masumoto.

**Writing – original draft:** Takahiro Sogawa, Daisuke Nagasato.

**Writing – review & editing:** Daisuke Nagasato, Yoshinori Mitamura.

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
