## [Decision Letter · Decision Letter 0]

20 Jan 2020

PONE-D-19-34572

Accuracy of a deep convolutional neural network in the detection of myopic macular diseases using swept-source optical coherence tomography

PLOS ONE

Dear Dr. Nagasato,

Thank you for submitting your manuscript to PLOS ONE. After careful consideration, we feel that it has merit but does not fully meet PLOS ONE’s publication criteria as it currently stands. Therefore, we invite you to submit a revised version of the manuscript that addresses the points raised during the review process.

The reviewer made several important comments that we would like to see addressed. I suggest that in addition to the current cohort used in DL analysis and that was presented intros MS, you add a group of patients with comorbidities that mimic what is encountered in real life and sand see how well the algorithm fares and present these data as well

We would appreciate receiving your revised manuscript by Mar 05 2020 11:59PM. To enhance the reproducibility of your results, we recommend that if applicable you deposit your laboratory protocols in protocols.io, where a protocol can be assigned its own identifier (DOI) such that it can be cited independently in the future. For instructions see: http://journals.plos.org/plosone/s/submission-guidelines#loc-laboratory-protocols

We look forward to receiving your revised manuscript.

Kind regards,

Demetrios G. Vavvas

Academic Editor

PLOS ONE

Journal Requirements:

2. Please include in your financial disclosure statement the name of the funders of this study (as well as grant numbers if available). If your study was unfunded, please revise your financial disclosure statement to “The author(s) received no specific funding for this work.

a) Please provide an amended Funding Statement that declares *all* the funding or sources of support received during this specific study (whether external or internal to your organization) as detailed online in our guide for authors at http://journals.plos.org/plosone/s/submit-now.  

b) Please state what role the funders took in the study.  If any authors received a salary from any of your funders, please state which authors and which funder. If the funders had no role, please state: "The funders had no role in study design, data collection and analysis, decision to publish, or preparation of the manuscript."

Reviewers' comments:

Reviewer's Responses to Questions

**Comments to the Author**

1. Is the manuscript technically sound, and do the data support the conclusions?

Reviewer #1: Yes

2. Has the statistical analysis been performed appropriately and rigorously? 

Reviewer #1: Yes

3. Have the authors made all data underlying the findings in their manuscript fully available?

Reviewer #1: Yes

4. Is the manuscript presented in an intelligible fashion and written in standard English?

Reviewer #1: Yes

5. Review Comments to the Author

Reviewer #1: 1. Finacial Disclosure is missing:Although it is implied that this study was funded as per the phrase 'the funders had no role in stud design, decision to publish or interpretation of the manuscript', the funding source is not mentioned.

2. The use of heatmaps to show areas of SS-OCT that the neural network focused compared to where retina specialists focused is very interesting.

3. Although the results of this study seem promising for Depp Learning, significant limitations still apply such as:

A. As mentioned in the manuscript, 'the original SS-OCT image size was 1,038 x 802 pixels but it was resized to 256 x 192 pixels to shorten the analysis time'

B. The exclusion of eyes with any other ocular coomrbidities including very common ones such as lens opacification, that favours DL as opposed to Retina Specialsts. The major limitation of this study is the design

of DL in evaluating a single condition at a time, excluding eyes with coexisting retinal diseases. Improvement in this aspect is needed before DL could potentially be used in real life clinical settings.

C. It would be interesting to train a DL network pass the limited binary or ternary classification and compaaae results accuracy with that of human Ophthalmologists.

4. In the discussion, it is mentioned the ' in the present study, the DL model we used was able to obtain the same correct answer rate in a shorter time relative to Ophthalmologists using the same sample data', yet both in methods and results of the present study such comparisons of time are missing.

5. In the baseline characteristics, AL and sex may be unavoidable to a certain extent in this type of study, yet the groups could have been matched in terms of age. There are significant differences in the age among the 4 groups of the present study p<0.001

6. line 133: please correct 'itthe'

6. PLOS authors have the option to publish the peer review history of their article (what does this mean?). If published, this will include your full peer review and any attached files.

Reviewer #1: No

---

## [Author Response · Author response to Decision Letter 0]

4 Mar 2020

We appreciate the careful reviews and instructive suggestions from the reviewers. We have revised our manuscript following your suggestions. In the course of doing so, we discovered incorrect word in Table 4. And we have added patients with comorbidities (mild cat, epiretinal membrane, chorioretinal atrophy, macular hole and so on) to the nHM and HM groups that mimic those encountered in real life, as you and the reviewer kindly pointed out. And the sentence “Some nHM and HM images included with comorbidities (mild cat, epiretinal membrane, macular hole and so on).” has been added (page7, lines100-102). Also, the sentences “Third, the DL performed in this study, though it showed high classification performance results, was limited to yielding only three types of classification. However, HM is often associated with various complications such as chorioretinal atrophy and macular hole retinal detachment. Therefore, it is necessary to train a network using a large dataset of HM fundus images with other complications.”have been deleted (Page 24, Line 368). Furthermore, as you advised, we added comorbidities images and re-analyzed. Therefore, the result data has changed slightly (All are highlighted).

We hope you will accept these changes. Our responses to the editor and reviewers are as follows:

Journal Requirements:

Reply: Thank you for your suggestion. We confirmed that our revised manuscript meets PLOS ONE’s style.

2. Please include in your financial disclosure statement the name of the funders of this study (as well as grant numbers if available). If your study was unfunded, please revise your financial disclosure statement to “The author(s) received no specific funding for this work.

Reply: Thank you for your suggestion. The authors received no specific funding for this work. As your pointed out, the sentence “The authors received no specific funding for this work” has been added in the financial disclosure section of the cover letter.

a) Please provide an amended Funding Statement that declares *all* the funding or sources of support received during this specific study (whether external or internal to your organization) as detailed online in our guide for authors at http://journals.plos.org/plosone/s/submit-now. 

b) Please state what role the funders took in the study. If any authors received a salary from any of your funders, please state which authors and which funder. If the funders had no role, please state: "The funders had no role in study design, data collection and analysis, decision to publish, or preparation of the manuscript."

Reply: Thank you for your helpful suggestion. The authors received no specific funding for this work. As your pointed out, the sentence “The funders had no role in study design, data collection and analysis, decision to publish, or preparation of the manuscript.” has been added in the financial disclosure section of the cover letter. 

Reviewers' comments:

Review Comments to the Author

Reviewer #1: 1. Finacial Disclosure is missing:Although it is implied that this study was funded as per the phrase 'the funders had no role in stud design, decision to publish or interpretation of the manuscript', the funding source is not mentioned.

Reply: Thank you for your helpful suggestion. As your and our editor pointed out, the sentences “The authors received no specific funding for this work. The funders had no role in study design, data collection and analysis, decision to publish, or preparation of the manuscript.” have been added in the financial disclosure section of the cover letter.

2. The use of heatmaps to show areas of SS-OCT that the neural network focused compared to where retina specialists focused is very interesting.

Reply: Thank you for your comment. We think heat maps is very helpful to know how AI diagnosis.

3. Although the results of this study seem promising for Depp Learning, significant limitations still apply such as:

A. As mentioned in the manuscript, 'the original SS-OCT image size was 1,038 x 802 pixels but it was resized to 256 x 192 pixels to shorten the analysis time'

Reply: We agree your suggestion. Your suggestion is the objective limitation of our study. As your pointed out, the sentence “Third, to shorten the analysis time, the original SS-OCT image with 1,038 × 802 pixels was resized to 256 × 192 pixels.” has been added (page24, lines 368-369).

B. The exclusion of eyes with any other ocular coomrbidities including very common ones such as lens opacification, that favours DL as opposed to Retina Specialsts. The major limitation of this study is the design of DL in evaluating a single condition at a time, excluding eyes with coexisting retinal diseases. Improvement in this aspect is needed before DL could potentially be used in real life clinical settings.

Reply: Thank you for your helpful suggestions. As your pointed out, we have added patients with comorbidities (mild cat, epiretinal membrane, chorioretinal atrophy, macular hole and so on) to the nHM and HM groups that mimic those encountered in real life. And we analyzed again including additional images. Thus, we rewrote Table 1. And the sentence “Some nHM and HM images included with comorbidities (mild cat, epiretinal membrane, chorioretinal atrophy, macular hole and so on). ”has been added (page7, line100-102).

C. It would be interesting to train a DL network pass the limited binary or ternary classification and compare results accuracy with that of human Ophthalmologists.

Reply: We are glad you are interested in our analysis method in this study.

4. In the discussion, it is mentioned the ' in the present study, the DL model we used was able to obtain the same correct answer rate in a shorter time relative to Ophthalmologists using the same sample data', yet both in methods and results of the present study such comparisons of time are missing.

Reply: Thank you for your suggestion. This is our mistake. According to this comment, the sentence “in a shorter time” has been deleted (Page 23, Line 348).

5. In the baseline characteristics, AL and sex may be unavoidable to a certain extent in this type of study, yet the groups could have been matched in terms of age. There are significant differences in the age among the 4 groups of the present study p<0.001

Reply: Thank you for your helpful comment. We agree that age adjustment is necessary. However, the patients with retinoschisis and mCNV increase with age, and the age could not be adjusted to secure a sufficient number of cases. If we can increase the number of target patients like a conducting joint research with other institutions, we think that we can adjust the age and lead to high-quality results. Please let us consider it for future study.

6. line 133: please correct 'itthe'

Reply: Thank you for your comment. the word “itthe” have been changed to “it the” (Page 9, Lines 134).

---

## [Decision Letter · Decision Letter 1]

19 Mar 2020

PONE-D-19-34572R1

Accuracy of a deep convolutional neural network in the detection of myopic macular diseases using swept-source optical coherence tomography

PLOS ONE

Dear Dr. Nagasato,

Thank you for submitting your manuscript to PLOS ONE. After careful consideration, we feel that it has merit but does not fully meet PLOS ONE’s publication criteria as it currently stands. Therefore, we invite you to submit a revised version of the manuscript that addresses the points raised during the review process.

Please adjust your limitations sections appropriately. We look forward ot the revised version 

We would appreciate receiving your revised manuscript by May 03 2020 11:59PM. To enhance the reproducibility of your results, we recommend that if applicable you deposit your laboratory protocols in protocols.io, where a protocol can be assigned its own identifier (DOI) such that it can be cited independently in the future. For instructions see: http://journals.plos.org/plosone/s/submission-guidelines#loc-laboratory-protocols

We look forward to receiving your revised manuscript.

Kind regards,

Demetrios G. Vavvas

Academic Editor

PLOS ONE

Reviewers' comments:

Reviewer's Responses to Questions

**Comments to the Author**

1. If the authors have adequately addressed your comments raised in a previous round of review and you feel that this manuscript is now acceptable for publication, you may indicate that here to bypass the “Comments to the Author” section, enter your conflict of interest statement in the “Confidential to Editor” section, and submit your "Accept" recommendation.

Reviewer #1: All comments have been addressed

2. Is the manuscript technically sound, and do the data support the conclusions?

Reviewer #1: Yes

3. Has the statistical analysis been performed appropriately and rigorously? 

Reviewer #1: Yes

4. Have the authors made all data underlying the findings in their manuscript fully available?

Reviewer #1: Yes

5. Is the manuscript presented in an intelligible fashion and written in standard English?

Reviewer #1: Yes

6. Review Comments to the Author

Reviewer #1: One additional comment:

In Machine Learning and Deep Learning, a major concern and current limitation of AI algorithms created via training sets obtained using only one imaging device is the extent of generalizability to other commercially available similar imaging devices. In this study, the Topcon DRI OCT-1 was used for all OCT images. The authors should state in their limitations section that the AI algorithms created and tested herein might not be generelizable to other commercially available similar imaging devices - the extent of generalizability across different devices is yet to be investigated.

Lines 311-314: the respective reference needs to be cited in the manuscript text.

7. PLOS authors have the option to publish the peer review history of their article (what does this mean?). If published, this will include your full peer review and any attached files.

Reviewer #1: No

---

## [Author Response · Author response to Decision Letter 1]

23 Mar 2020

Editor,

PLoS ONE

March 22, 2020

Re: PONE-D-19-34572R1 entitled " Accuracy of a deep convolutional neural network in the detection of myopic macular diseases using swept-source optical coherence tomography".

To the Editor,

Thank you for your letter of March 20th, 2020 and for sending us the referee’s comments on our manuscript. We are returning the manuscript revised according to the comments.

We have studied the comments carefully and have made corrections which we hope meet with your approval. In the revised manuscript, all changes have been highlighted in yellow. Each of the coauthors has seen and agreed with each of the changes made to this manuscript in the revision.

---

## [Editor Report · Decision Letter 2]

30 Mar 2020

Accuracy of a deep convolutional neural network in the detection of myopic macular diseases using swept-source optical coherence tomography

PONE-D-19-34572R2

Dear Dr. Nagasato,

We are pleased to inform you that your manuscript has been judged scientifically suitable for publication and will be formally accepted for publication once it complies with all outstanding technical requirements.

With kind regards,

Demetrios G. Vavvas

Academic Editor

PLOS ONE
---

## [Editor Report · Acceptance letter]

2 Apr 2020

PONE-D-19-34572R2 

Accuracy of a deep convolutional neural network in the detection of myopic macular diseases using swept-source optical coherence tomography 

Dear Dr. Nagasato:

I am pleased to inform you that your manuscript has been deemed suitable for publication in PLOS ONE. Congratulations! Your manuscript is now with our production department. 

With kind regards,

on behalf of

Dr. Demetrios G. Vavvas 

Academic Editor

PLOS ONE